# The Effect of Resistance Training with Outdoor Fitness Equipment on the Body Composition, Physical Fitness, and Physical Health of Middle-Aged and Older Adults: A Randomized Controlled Trial

**DOI:** 10.3390/healthcare12070726

**Published:** 2024-03-26

**Authors:** Pablo J. Marcos-Pardo, Alejandro Espeso-García, Raquel Vaquero-Cristóbal, Tomás Abelleira-Lamela, Noelia González-Gálvez

**Affiliations:** 1Centro de Investigación para el Bienestar y la Inclusión Social (CIBIS Research Center), SPORT Research Group (CTS-1024), Department of Education, Faculty of Education Sciences, University of Almería, 04120 Almería, Spain; pjmarcos@ual.es; 2Active Aging, Exercise and Health/HEALTHY-AGE Network, Consejo Superior de Deportes (CSD), Ministry of Culture and Sport of Spain, 28040 Madrid, Spain; ngonzalez@ucam.edu; 3Injury Prevention in Sport Research Group (PRELEDE), Facultad de Deporte, UCAM Universidad Católica de Murcia, 30107 Murcia, Spain; tabelleira@ucam.edu; 4Department of Physical Activity and Sport Sciences, Faculty of Sports Sciences, University of Murcia, 30720 Murcia, Spain; 5Research Group on Health, Physical Activity, Fitness and Motor Behaviour (GISAFFCOM), Facultad de Deporte, UCAM Universidad Católica de Murcia, 30107 Murcia, Spain

**Keywords:** aging, healthy aging, outdoor gym

## Abstract

This study examined the effect of outdoor-fitness-equipment-based resistance training on the health parameters of middle-aged and older adults, as well as analyzing the effect of age on the results found. A total of 149 volunteers were randomly assigned to the training (TG) and control (CG) groups. The TG performed two weekly sessions of resistance training for 8 weeks using outdoor fitness equipment, while the CG continued with their regular daily activities. Body composition was measured using DXA, and the maximal isometric voluntary contraction in knee extension, elbow flexion, and hand grip were assessed, along with the 4 m walk test, the Timed Up and Go Test, and the Short Form 36 Health Survey Questionnaire. The TG showed a significant increase in the lean mass index (*p* = 0.002) and maximal isometric voluntary contraction in both legs (*p* < 0.001) and arms (*p* < 0.001), as well as in physical functioning (*p* < 0.001) and the role physical dimension (*p* = 0.006) of the Short Form 36 Health Survey Questionnaire, compared to the CG, which showed a decrease in all these variables. In addition, the TG showed a greater decrease in fat mass (*p* < 0.001), fat mass index (*p* = 0.003), and the Timed Up and Go Test (*p* < 0.001) than the CG. Age conditioned the evolution of most of the variables analyzed (*p* < 0.05). In conclusion, resistance training with outdoor fitness equipment may be useful for improving the health of middle-aged and older adults, although age is a factor that could influence the adaptations found.

## 1. Introduction

In an era where the emphasis on health and wellness has gained paramount importance, understanding the profound impact of physical activity on middle-aged and older adults has become increasingly crucial [1]. As age advances, optimizing body composition and enhancing physical fitness are essential components of sustaining an active and fulfilling lifestyle [2].

According to the World Health Organization (WHO), moderate to vigorous physical activity can reduce the likelihood of middle-aged and older people developing chronic diseases such as heart disease, stroke, diabetes, sarcopenia, and certain types of cancer by approximately 30–40% [1]. Also, physical activity allows older adults to reduce their risk of mobility limitations and decreased quality of life [3]. Moreover, physical activity can attenuate changes in body composition, such as increased body fat and decreased lean muscle mass, which can contribute to a range of health issues, including metabolic disorders and reduced functional capacity [4].

Within the different modalities of physical activity, researchers and fitness experts have turned to progressive resistance training as a potential solution. Maintaining a regular commitment to resistance training regimens stands as a notable element in the aging journey, potentially functioning as both a proactive strategy and a therapeutic choice for a range of musculoskeletal conditions [5]. This form of exercise involves gradually increasing the resistance during workouts, thereby stimulating muscle growth, promoting overall strength, and helping to improve the functional capacity, muscle mass, and quality of life of older adults [6].

The importance of strength training in promoting healthy aging and improving the quality of life of adults and older adults has been widely documented [2]. It has been shown that training at least two days a week at moderate to vigorous intensity can have significant effects on body composition, physical fitness, physical health, and overall physical well-being [7]. Traditionally, strength training in middle and older adults has focused on self-weight bearing or the use of equipment such as resistance bands and on activities guided by monitors and performed mainly in specialized centers or indoor environments [2,8]. However, although these methods have been shown to be effective in improving health and physical function, this approach may present barriers in terms of accessibility and cost, which may limit their widespread adoption [8].

In this sense, public areas are ideal for encouraging physical activity and resistance training, as they encompass both green spaces and structures that are purpose-built to foster physical activity [9]. More specifically, outdoor fitness equipment (OFE) refers to facilities or exercise equipment, adapted from traditional gym configurations, specially designed for outdoor use, which generally use the individual’s own body weight for resistance [10,11]. OFE emerges as an innovative and accessible opportunity to integrate strength training into the daily lives of this population [9,12]. These free-to-use facilities, located in public spaces around the world, feature guided machines with clear instructions for use [13]. This allows anyone, regardless of their prior knowledge or experience, to engage in physical activity. In addition, the outdoor and accessible nature of these environments encourages interaction and mutual support among users, enhancing their exercise experience [14,15].

However, despite the growing amount of OFE in parks, studies on its specific effects on the health and well-being of adults and older adults are limited [16,17]. Indeed, a recent narrative review suggests that while OFE may have the potential to improve the health of users, there is a great lack of rigorous experimental studies demonstrating its actual effects in the medium to long term [14]. In line with this, most of the previous research has been based on a small number of machines or on a small sample size or has not focused on health-related variables [16,18]. Moreover, the design of OFE machines, mainly leveraging the user’s body weight as unique external charge, raises considerations about their effectiveness in achieving the minimum intensity levels for significant health benefits and long-term training progression [11]. Furthermore, while the potential of OFE to deliver effective fitness solutions deserves exploration, its distinct nature from traditional fitness centers, and in particular its reliance on the user’s body weight, warrants a tailored investigation to ascertain its true efficacy.

In view of the above, this current study addresses these gaps. More specifically, the aims were to assess the impact of a resistance training program conducted in OFE on health-related variables, including the body composition, physical fitness, and physical health of middle-aged and older adults, as well as analyzing the effect of age on the results found.

## 2. Materials and Methods

### 2.1. Experimental Approach to the Problem

The present study is an 8-week two-armed parallel single-blind randomized controlled trial with blinded examiners, which follows the methodology described in a protocol study already published in a previous research work [19]. The participants were divided equally and randomly between the training group (TG) and the control group (CG). The trial design was registered with ClinicalTrials.gov (identifier: NCT04958499) and followed the Consolidated Standards of Reporting Trials (CONSORT) guidelines and Standard Protocol Items: Recommendations for Interventional Trials statements [20,21].

### 2.2. Subjects

Participants were recruited on a voluntary basis through informative conferences and advertising in senior centers located in the municipalities surrounding the university facilities in Murcia, Spain. The inclusion criteria were (a) being over 50 years of age; (b) being physically independent; (c) not performing resistance training or physical exercise systematically for at least 1 year before. The exclusion criteria were (a) self-reported substance abuse or dependence disorders; (b) suffering from any type of injury or pathology that prevented the performance of the measurements and/or training; (c) being medically prescribed to take medications that could have influenced physical performance; (d) having undergone surgery that would have prevented performing any of the tests and/or training or posed a risk. Moreover, further exclusion criteria were included for both groups: (a) absence in the post-test evaluations; (b) the initiation of other systematic physical exercise during the study period and for the TG participants specifically; (c) absence from more than 25% of the intervention program sessions. The final sample was composed of 128 middle aged and older adults, of which 28 were middle-aged men (age = 56.57 ± 3.91 years old, max = 64, min = 50; height = 169.17 ± 6.92 cm, max = 181.2, min = 158.1; body mass = 82.70 ± 9.82 kg, max = 106.60, min = 66.70; and BMI = 29.00 ± 3.91 kg/m^2^, max = 37.90, min = 22.68); 18 were older adult men (age = 66.61 ± 2.17 years old, max = 72, min = 65; height = 166.28 ± 8.59 cm, max = 174.50, min = 154.70; body mass = 77.96 ± 15.21 kg, max = 127.70, min = 64.10; and BMI = 28.18 ± 5.00 kg/m^2^, max = 44.76, min = 24.04); 62 were middle-aged women (age = 54.58 ± 4.82 years old, max = 63, min = 50; height = 157.38 ± 6.25 cm, max = 180.40, min = 148.00; body mass = 63.87 ± 8.90 kg, max = 96.00, min = 45.70; BMI = 25.79 ± 3.23 kg/m^2^, max = 33.61, min = 19.17); and 20 were older adult women (age = 68.75 ± 3.61 years old, max = 77, min = 65); height = 153.97 ± 3.98 cm, max = 159.00, min = 146.50; body mass = 67.94 ± 8.63 kg, max = 89.10, min = 55.20; and BMI = 28.72 ± 4.06 kg/m^2^, max = 37.83, min = 22.62). Subsequently, the participants were randomized assigned to the TG (*n* = 64; 29 males and 35 females; age = 60.02 ± 7.25 years old; height = 163.96 ± 9.30 cm; body mass = 75.42 ± 15.06 kg; and BMI = 27.97 ± 4.77 kg/m^2^) and CG (n = 64; 17 males and 47 females; age = 57.83 ± 6.92 years old; height = 157.40 ± 6.37 cm; body mass = 65.79 ± 7.50 kg; and BMI = 26.60 ± 2.99 kg/m^2^). The CONSORT flow diagram is shown in Figure 1.

### 2.3. Randomization and Blinsding

Following recruitment and selection, the participants were randomized using a computer-generated random number table, ensuring that allocation was undertaken impartially. The chosen ratio for the randomized groups was 1:1. This process ensured that each participant had the same chance of being assigned to either of the two groups, removing any selection bias. Furthermore, a single-blind study design was adopted based on previous studies [22], where the evaluators were blinded and unaware of the group assignments, as well as the individuals’ results for previous measurements, which helps to avoid possible biases and improve the objectivity of the study findings.

### 2.4. Procedures

The TG participants engaged in an 8-week training program utilizing OFE, with two weekly sessions lasting one hour per session on non-consecutive days. The CG participants did not perform any training and were asked to maintain their usual lifestyle [19]. Attendance at the training sessions was monitored daily, with an average rate of 92.75%. Given this high attendance rate, only 6 participants were excluded from the study due to a lack of adherence to the training program.

The intervention program was carried out in an OFE circuit, consisting of 11 exercises performed on eight body weight OFE (Bonny rider, Air Walker, Surfboard, Row, Parallel bars, Gemini, Flyer wheels, and Swing) provided by the company Equipamientos para Entorno Urbano S.L. (Murcia, Spain) (Table 1), located in a private area annexed to the laboratory within the facilities of the Catholic University of Murcia (UCAM) [19].

The training program was designed and concurrently supervised by two graduates in Sports Science with master’s degrees in Strength and Conditioning Training, Healthy Aging, and Physical Activity and Health. Each resistance training session was preceded by a standardized warm-up routine, focused on joint mobility, cardiovascular, and body weight exercises, following a previous protocol [19]. The training protocol utilized is detailed in Table 2, which outlines the progression of the training over the 8 weeks of the intervention in terms of its volume and intensity. The circuit is based on the work time for each exercise and each set (T of work/exe), which indicates the exercise duration for each machine and each set, and the rest time between exercises (T of rest/exe), which denotes the rest interval between machines within the same set. The execution speed measured in seconds for the concentric and eccentric phases is also included (concentric/eccentric), as well as the total number of repetitions participants had time to carry out for each exercise in each set (Reps exe). In addition, the table includes information on the rest pause between sets or circuit rounds (T rest/set), the total duration of the session in minutes (T session), and the total volume of repetitions per session (Total Reps/session). The training load was regulated by the muscle time under tension using a digital metronome to set the execution rhythm of the concentric and eccentric phases for each piece of OFE [23]. This approach ensured that all the repetitions were performed at a controlled and constant speed, with no variations in the execution velocity. To achieve this, the two program supervisors provided individual and continuous feedback to each participant. After each session, a cool-down period was provided, consisting of a series of static active stretches of each of the muscle groups involved in the training [19].

The measurements were performed in a sports science laboratory under standardized conditions. Firstly, the participants completed a sociodemographic questionnaire that was created ad hoc for this study, which included questions about age, gender, physical activity level, illness, and injuries. Furthermore, the dimensions of physical functioning and role physical from the Spanish validated version of the Short Form 36 Health Survey Questionnaire (SF-36) were used to measure health status [24,25,26]. The participants completed the ad hoc and SF-36 questionnaires anonymously and individually as hard copies. The participants did not receive any additional explanation about the purpose of the questionnaires, other than that contained in the questionnaire itself.

After completing the questionnaires, body mass and height were measured according to the protocol from the International Society for the Advancement of Anthropometry (ISAK) [27], performed by an ISAK Level 1 accredited anthropometrist and in accordance with previous studies [19]. Subsequently, body composition was determined using dual-energy X-ray absorptiometry (DXA) following the protocol used previously [19,28], which was used for the measurement of fat mass (kg), fat mass index (kg/m^2^), lean mass (kg), and lean mass index (kg/m^2^).

This was followed by a warm-up similar to that of the training sessions, as previously described [29]. After this, a maximum voluntary isometric contraction (MVIC) leg extension test, a MVIC elbow flexion test, a 4 m gait walk, and a Timed Up and Go test (TUG) were performed in a randomized order, following a previous protocol [19]. Before starting the MVIC tests, familiarization was performed, after which a 5 min rest period was allowed before two non-consecutive maximal attempts were performed, with 2 min of rest between tests. The maximum peak force in Newtons (N) between the two attempts of each test was recorded for the MVIC test, and the best time was registered for the 4 m and TUG tests.

### 2.5. Statistical Analyses

Based on previous studies, groups larger than 30 were required for the results to reliably characterize precision errors or changes during clinical monitoring [30]. A dropout rate of 16% was assumed based on previous studies [31], so a minimum of 35 participants had to be included per group. The RStudio 3.15.0 (Rstudio Inc., Boston, MA, USA) software was used for calculation of the sample size. The sample size and statistical power were established considering the standard deviation (167.46 N) in the maximum voluntary isometric contraction (MVIC) leg extension test [32]. A significance power of 95% (1−β = 0.95) and a significance level of 0.05 were considered. Ultimately, the sample consisted of 64 participants, resulting in an assumed error of 41.02 N. Due to the use of a convenience sampling method, a significantly larger sample than required was utilized since the recruitment of participants was conducted across several senior centers and many individuals enrolled together. This approach ensured a sufficient sample size in case of low adherence to the program, improving the generalizability and reliability of the results.

After analyzing the normality of the variables using the Kolmogorov–Smirnov test, as well as kurtosis, skewness, and variance, two-way ANOVAs with one-way repeated measures were carried out to analyze the inter- and intra-group differences. A two-way ANCOVA with one-way repeated measures was also performed to determine the influence of the covariate of age on the results obtained on the study variables. Partial eta squared (η^2^) was used to calculate the effect size and was defined as small: ES ≥ 0.10; moderate: ES ≥ 0.30; large: ≥1.2; or very large: ES ≥ 2.0, with an error of *p* < 0.05 [33]. A value of *p* < 0.05 was set to determine the statistical significance. The statistical analysis was performed using the SPSS statistical package (v. 25.0; SPSS Inc., Chicago, IL, USA).

## 3. Results

Prior to the beginning of this study, all the participants were inactive and had not undergone systematic resistance training in the last year, thus fulfilling the established inclusion and exclusion criteria. The sample included 9 individuals with controlled high blood pressure, 11 with controlled cholesterol levels, 2 with controlled diabetes, 8 with lumbar discomfort, 3 with asthma, and 4 with osteoporosis. These conditions were thoroughly reviewed by healthcare professionals and were determined to meet the inclusion criteria, as they were controlled and did not interfere with the participants’ ability to engage in the study’s activities or measurements. This allowed us to proceed with the training and measurements in a normal manner.

Table 3 shows the pre- and post-intervention differences in body composition, physical fitness, and physical health of the groups. In terms of the body composition variables, the TG showed a significant decrease in fat mass (*p* < 0.001, ES = 0.257) and the fat mass index (*p* < 0.001, ES = 0.221) and a significant increase in the lean mass index (*p* = 0.045, ES = 0.044). In contrast, the CG showed a significant decrease in lean mass (*p* = 0.024, ES = 0.012) and the lean mass index (*p* = 0.008, ES = 0.037) with a small effect size on all these variables. In terms of the physical fitness variables, the TG showed a significant increase in MVIC in both legs (*p* < 0.001) with a moderate effect size (ES = 0.207 for the right, ES = 0.154 for the left) and MVIC in the arms (*p* < 0.001, ES = 0.521) with a large effect size; the TUG test (*p* < 0.001, ES = 0.393) indicated a large effect size; and the 4 m gait walk (*p* = 0.006, ES = 0.057) showed a small effect size, whereas the CG showed a significant decrease in MVIC of the left leg (*p* = 0.019, ES = 0.051) and MVIC of the arms (*p* < 0.001, ES = 0.073), both with a small effect size, and a significant increase in the TUG test (*p* < 0.001, ES = 0.102) with a moderate effect size. As for the physical health results according to the SF-36 scores, the CG showed a significant decrease in the physical functioning and role physical dimensions.

The age-adjusted analysis revealed significant changes in the TG for fat mass (*p* < 0.001, ES = 0.263), the fat mass index (*p* < 0.001, ES = 0.230), and the lean mass index (*p* = 0.022, ES = 0.042), all showing a small effect size; for MVIC for both the right (*p* < 0.001, ES = 0.199) and left legs (*p* < 0.001, ES = 0.146), with a small effect size; for MVIC in the arms (*p* < 0.001, ES = 0.532), with a large effect size; for the TUG test (*p* < 0.001; ES = 0.391), with a moderate effect size; and for the 4 m gait walk (*p* = 0.014, ES = 0.047), with a small effect size. Age also influenced the within-group changes in the CG in the variables of the lean mass index (*p* = 0.037, ES = 0.035), MVIC left leg (*p* = 0.015, ES = 0.047), MVIC arms (*p* = 0.001, ES = 0.082), the TUG test (*p* < 0.001, ES = 0.104), PH physical functioning (*p* < 0.001, ES = 0.107), and PH role physical (*p* = 0.004, ES = 0.066), all with small effect sizes.

Table 4 compares the changes observed before and after the intervention between the TG and the CG. The pre-post change showed in the TG was significant compared to the pre-post change in the CG in the lean mass index, with a small effect size (*p* = 0.002, ES = 0.077); fat mass (*p* < 0.001, ES = 0.109) and the fat mass index (*p* = 0.003, ES = 0.069), both with small effect sizes; MVIC for both the legs and arms, with small effect sizes for the legs (*p* < 0.001, ES = 0.135 for the right leg and *p* < 0.001, ES = 0.178 for the left leg) and a moderate effect size for the arms (*p* < 0.001, ES = 0.469); the TUG test, with a small effect size (*p* < 0.001, ES = 0.097); as well as in the physical functioning (*p* < 0.001, ES = 0.089) and role physical dimensions (*p* = 0.006, ES = 0.058), both with small effect sizes. Age had a significant influence on the inter-group differences in the pre-post change for all the analyzed variables (*p* < 0.001–0.002; ES = 0.073–0.481), except for lean mass (*p* = 0.115, ES = 0.020) and the 4 m gait walk test (*p* = 0.219, ES = 0.012).

## 4. Discussion

The main aim of the present study was to analyze the effect of resistance training with OFE on health-related variables in middle-aged and older adults. The present investigation showed a significant increase in lean mass and a decrease in fat mass in the group that received resistance training with OFE, while the CG showed a decrease in lean mass and no change in fat mass. Only two studies have applied intervention programs with OFE, both in older adults. One of them applied an intervention program that included seven types of OFE (two cardiovascular training, two strength, two mobility, and two balance machines), three sets, 8–12 repetitions, and 12 weeks of training [16], while the other study applied a 6-week program that included five types of OFE (three strength and two cardiovascular training machines), three sets, and 12–15 repetitions [18]. The study by Chow [16] showed no changes in the body composition variables, but as the study did not include a control group, no comparisons could be made with it. The results found in the present study may be due to the fact that it included middle-aged adults and not only older adults. Some research has shown different effects according to age, with significant changes observed in the body composition of middle-aged adults, while older adults did not show significant changes in body composition [34], or a larger effect size in middle-aged adults than in older adults [35], suggesting that OFE training may improve body composition in middle-aged adults and slow age-associated changes in body composition in older adults.

In addition, the difference between the results could also be due to the different prescribed volumes of the programs. Both Chow’s [16] study and Kim’s [18] study included a much lower number of strength exercises (2–3, respectively, vs. 11 exercises in the present study). Research studies on resistance training usually utilize at least six exercises [36], and a minimum dose has been identified as necessary to produce adaptations, ranging from low-volume and high-intensity to high-volume and low-intensity. However, resistance training with OFE is considered to be low- to moderate-intensity resistance training [11], and it is therefore considered necessary to utilize a sufficient volume in order to achieve body adaptations [14]. More research is needed to determine what the minimum volume of OFE training should be to produce adaptations.

Most of the research examining the effect of resistance training programs on body composition has been focused on middle-aged adults [36], older adults [37], overweight and obese people [35], people with sarcopenia [6], people with sarcopenia and obesity [38], or postmenopausal women [34]. Although all of them have shown improvements in most of the variables related to body composition, some of them do not show improvements in lean mass [6]. This could be due to the dose applied in resistance training.

Another objective of the present study was to analyze the effect of resistance training with OFE on the physical fitness of middle-aged and older adults. Improvements were observed in upper and lower limb strength, as well as in the Timed Up and Go test and gait speed. Resistance training programs have shown improvements in muscle strength and muscle function [6] or show a trend towards improvement, as well as improvements in strength and muscle performance [6]. In this sense, this shows that neural mechanisms and muscle innervation, such as adaptations in activation, timing, and rhythm coding, and not muscle hypertrophy, are the most likely reasons for the increase in muscle strength, mainly in beginners during the first 8 weeks [39]. It has been reported that improvements in muscle performance are mainly associated with an increase in muscle strength rather than muscle mass after resistance training. In this sense, resistance training with OFE could be of added benefit to resistance training using other machines or other equipment given the reports of an improvement not only in muscle strength and function but also an increase in muscle mass in the first 8 weeks [40].

Resistance training has been shown to change skeletal muscle properties, leading to an increase in lean body mass and a reduction in exercise-induced oxidative stress [41]. The main reason for the increase in lean mass is the increase in muscle mass, which may be due to intramuscular anabolic signaling, the maximization of the response of muscle fiber recruitment, the time under tension, or the metabolic stress [42]. Some research indicates that there seems to be evidence that moderate–high loads (>60% 1RM) and lower–moderate volumes (<12 repetitions) produce a better response in muscle mass, although in both cases (moderate–high vs. low loads), adaptations have been observed [6]. A repetition continuum or strength–endurance continuum proposes that the adaptations produced will depend on the number of repetitions and the intensity of the load [43]. However, it appears that muscle mass improvements can be achieved with loads greater than 30% 1RM, indicating that there is no ideal zone from a practical point of view [44]. In contrast, in relation to the effect on fat mass, there seems to be more evidence about the need for low to moderate intensity and high volumes [35]. A possible explanation could be the relationship between lipolysis with low intensity training and some volume because the optimal intensity for fat oxidation has been described as around 60% [45]. In this sense, it has been described how resistance training programs of low–moderate intensity and moderate–high volume are the most suitable when the objective is both to reduce fat mass and increase muscle mass without changes in body mass [36]. The present program utilizes between 12 and 15 repetitions, and it is therefore considered moderate-intensity, as well as optimal training for both a muscle mass increase and fat mass reduction.

In relation to the concept “strength–endurance continuum”, it is considered that the adaptations produced by light load protocols are specific to endurance muscle adaptations and that the ability to improve maximal strength is minimal [43]. This may be because studies that have used 1RM testing do support the repetition continuum with respect to muscle strength in dynamic constant resistance exercise, and studies that used a device to assess isometric strength showed no difference, which may be due to the specificity of the test with respect to training [44].

Both previous studies discussed and the present research have shown improvements in strength using moderate loads [6,37,38]. Early adaptations to strength training in strength improvement occur through motor learning and neural enhancement, and it is possible for any load to influence strength. However, maximizing the strength-related results requires heavier loads for experienced individuals [46]. In this sense, the effect of OFE should also be investigated in experienced individuals.

OFE resistance training involves moderate intensities and a number of repetitions. These intensities are interesting for the sedentary population, because, as described, they produce improvements in body composition, strength, and muscle function. Furthermore, it seems reasonable to prescribe moderate loads given that training with light loads involves performing many repetitions, increases the training time, and produces high levels of metabolic acidosis, which tends to cause discomfort, affecting adherence. On the other hand, the minimum dose approach to resistance training minimizes the barriers to physical activity practice, favoring adherence and reducing negative affective responses, such as increased discomfort and reduced satisfaction [17].

Another aim of the present study was to analyze the effect of resistance training with OFE on the physical health and quality of life of middle-aged and older adults. Previous meta-analyses have reported improvements in quality of life, specifically in the physical domains, after resistance training [47]. These quality of life domains have been influenced by aspects such as disability, diseases, and socioeconomic status; thus, these domains decrease with age [48]. In this sense, the present research showed no changes in physical functioning, although improvements were observed in the role physical dimension. At the same time, a decline was observed in the group that did not train (CG). This could be due to the fact that in our study, the volunteers were relatively healthy and had high values in these domains. Previous research has indicated that quality of life improved with training in older adults with various chronic conditions as compared to healthy controls [3] and those with low quality of life values prior to the intervention [40].

Likewise, improvements or decreases in deterioration could be related to improvements in muscle mass, strength, and muscle function, as well as an improvement in energy and the ability to perform the activities of daily living [49]. Therefore, the present research suggests that training with OFE could prevent a decline in these domains of quality of life in healthy middle-aged and older age adults.

Another important objective of the present study was to analyze the influence of age on the findings of the present research. It was found that age had a significant influence on the intergroup changes for almost all the variables analyzed, as well as on the differences found when comparing the changes in the TG and CG for most of the variables analyzed. Previous research has suggested that age may influence body composition, showing an increase in fat mass and a decrease in muscle mass with increasing age [50,51], which may be related to the appearance of numerous chronic diseases [52]. Other research has found that strength, resistance, and power decrease during the aging period [53], which could result in reduced functionality, autonomy, and quality of life [54,55]. Moreover, investigations reveal that as individuals age, there is a decline in both speed and agility [56,57], factors also associated with functionality and autonomy [58,59]. Indeed, the combination of losses of muscle mass, strength, and physical fitness that occur during the aging period has been referred to as sarcopenia [60], being a pathology that affects between 10 and 16% of older people [61]. Therefore, it is not surprising that the covariate of age showed a significant influence on the evolution of these variables of body composition and physical condition, including other more subjective variables, such as the participant’s perception of their physical functioning and role physical dimensions. In fact, previous research has shown that the effects of a circuit strength training program on body composition and fitness may be different depending on whether it is performed by middle-aged adults [62] or older people [63]. Therefore, although it can be indicated that this type of program is effective in both age groups, future research is needed to analyze the specific adaptations of the training to each age group.

The strength of the present study is its pioneering nature in demonstrating that an outdoor training circuit with OFE, consisting of 11 exercises performed on eight types of OFE, can induce adaptations in health-related variables (body composition, physical fitness, and physical health) in middle-aged and older adults, despite relying on the principle of body weight resistance. This finding is significant given that it enhances and expands the range of tools available for improving health and physical function in middle-aged and older adults, offering an accessible alternative that overcomes the traditional barriers associated with strength training, such as the need for expensive equipment or indoor facilities. Leveraging OFE placed in public spaces, this study not only highlights the possibility of effectively incorporating strength training into the lifestyle of this population group but also emphasizes the importance of inclusive and easily accessible training facilities that encourage the adoption of an active and healthy lifestyle.

While this study provides valuable insights, there are areas for further exploration that could enhance our understanding. One such area is the short duration of the intervention. Although improvements in some parameters were observed, it is conceivable that longer intervention durations could have had a greater impact on the results. Future studies must carry out interventions of longer durations to be able to provide long-term results. Another limitation could be the contaminating variable of nutrition. Nutrition was not monitored, and the volunteers were only given guidelines to maintain their nutritional habits. A further limitation concerns the quantification of the training load, which was performed using time under tension instead of traditional sets and repetitions since OFE does not allow for load adjustments. In addition, the lack of inclusion of a third group using traditional resistance training in the gym could be considered a limitation, as it would have been interesting to directly compare the effects of OFE training with conventional strength training methods. This approach could make it difficult to compare the results obtained with other studies that use different methods to quantify the training load. Another limitation was that the low number of participants prevented the separation of the TG and CG data between middle-aged and older adults. Given that the present research has found that age may influence the evolution of health-related variables in both TG and CG, this is an issue to be addressed in future studies. Furthermore, some of the participants had chronic illnesses or injuries, but the number of participants was not sufficient to analyze the influence of this issue on the evolution of the intervention. Future studies should aim to include a larger cohort of individuals with these medical backgrounds to facilitate a more comprehensive examination of the effects of training in these specific populations. Finally, attention must be paid to the limitation of the study regarding the replication of the equipment setup. It is important to note that if training is not conducted with the machines or training program used in this study, the results may not be extrapolatable. Finally, a possible limitation of the study is the small number of participants with pathologies or injuries, which precluded a detailed analysis of the effects of these conditions on the study variables due to their low incidence.

## 5. Conclusions

The present study found that resistance training with OFE twice a week for 8 weeks, despite only allowing self-loading, led to significant improvements in body composition, physical fitness, health, and quality of life among middle-aged and older adults compared to a control group that continued with their usual daily activities. The age of the participants influenced the evolution of the variables analyzed. The significant improvements observed through the use of OFE, based on the principle of body weight resistance, highlight an opportunity in the areas of clinical practice, public health, and physical fitness. This approach removes traditional barriers to resistance training, such as the costs of equipment and the need for indoor spaces, thereby democratizing health and fitness benefits. The detailed training program provided within this article ensures direct practical applicability and marks a fundamental breakthrough, promising a more inclusive future in health promotion and disease prevention strategies, making health improvements accessible to a wider population.

## 6. Patents

The results of this study were taken into consideration for the design of new OFE that will be more effective and safer, with the following registered patents: Publication n° ES1296848 (U) (http://invenes.oepm.es/InvenesWeb/detalle?referencia=U202231979) and ES1296869 (U) (http://invenes.oepm.es/InvenesWeb/detalle?referencia=U202231980) (accessed on 2 August 2023), as well as others that are currently under patent review.

## Figures and Tables

**Figure 1 healthcare-12-00726-f001:**
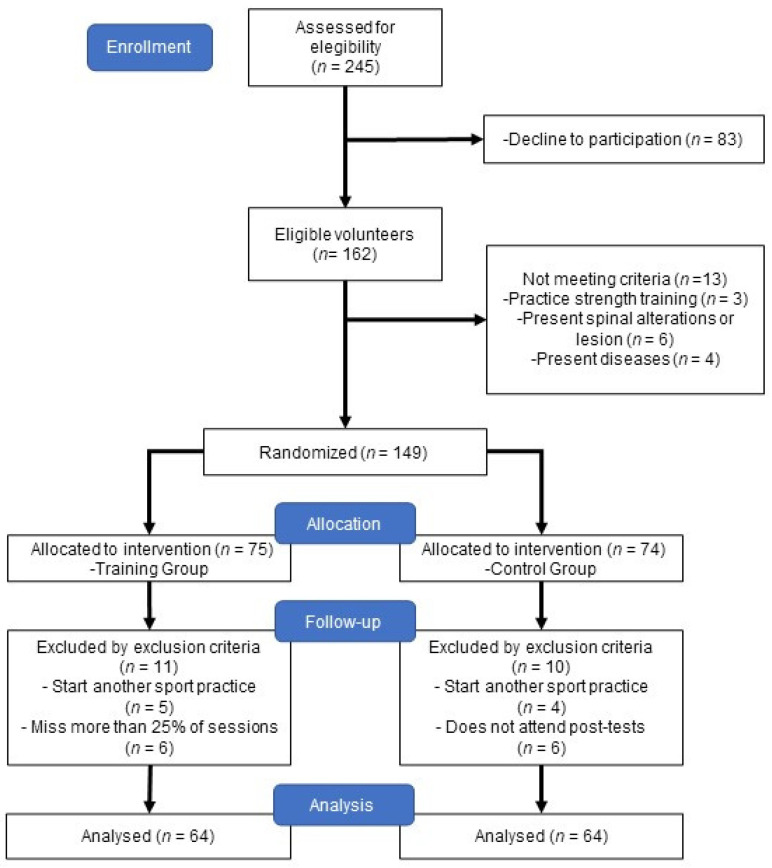
CONSORT flow diagram of participants through each stage of the study.

**Table 1 healthcare-12-00726-t001:** Exercises to be performed in the outdoor fitness equipment circuit.

Exercise	Method of Use	Representation
Bonny rider	Using a supine grip, pull the handle with both hands by flexing your elbows and extending your shoulders.	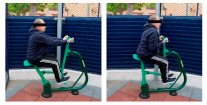
Shoulder press	On the Gemini machine, grasping the grips at a low height, push the handle by extending your elbows and flexing your shoulders.	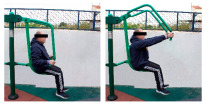
Air Walker	Holding onto the bar and standing on both platforms, perform hip flexion extensions, keeping the trunk perpendicular. Hold briefly and isometrically at the end of the range of motion.	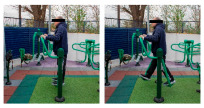
Tricep dips	On the parallel bars, holding with both hands and starting with your elbows flexed, extend them by overcoming the load of your own weight. Adaptation: triceps push-ups with a prone grip sideways to the machine.	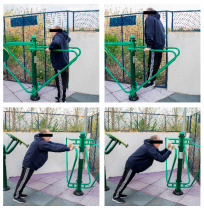
Shoulder wheel	On the Flyer wheels machine, with your elbows fully extended and grasping the grips, perform circumduction through shoulder mobility.	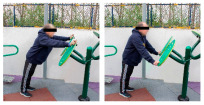
Chest press	On the Gemini machine, holding the grips at a middle height, push the handle by extending your elbows and flexing your shoulders.	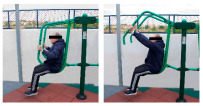
Rotator cuff wheel	With the Flyer wheels machine to your side, support your elbow and hold your grip. Perform internal and external rotation.	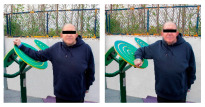
Calf raise	On the Swing machine, with your knees extended, rest your metatarsals on the lower edge of the platform. Flex and extend your ankles.	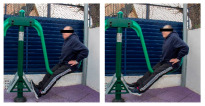
Surfboard	Standing on the platform and holding onto the handle, perform a pendulum movement from the waist in the frontal plane. Hold briefly and isometrically at the end of the range of motion.	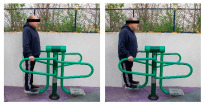
Leg press	On the Swing machine, while sitting with your feet on the platforms, push through knee extension to overcome the load.	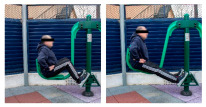
Row	Sitting on the seat and with your feet in position, pull on the grip by flexing your elbows and externally rotating your shoulders, keeping the trunk perpendicular to the ground.	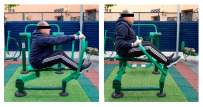

**Table 2 healthcare-12-00726-t002:** Training plan.

Week	Sets	T of Work/Exe (s)	Reps Exe	Concentric/Eccentric (Repetition/s)	T Rest/Exe (s)	T Rest/Set (min)	T Session (min)	Total Reps/Session
1	1	30	15	1/1	45	-	13.75	165
2	2	30	15	1/1	30	2	27	330
3	3	30	15	1/1	30	4	43	495
4	3	30	15	1/1	30	4	43	495
5	3	45	15	1/2	30	4	51.25	495
6	3	45	15	1/2	30	4	51.25	495
7	3	45	11	2/2	30	4	51.25	371
8	3	45	11	2/2	30	4	51.25	371

Exe: exercise; Reps: repetitions; T: time.

**Table 3 healthcare-12-00726-t003:** Intra-group changes in body composition, physical fitness, and physical health variables.

					Unadjusted			Adjusted by Age		
Variable	Group	Pre-Intervention (X ± SD)	Post-Intervention (X ± SD)	Mean Dif.	95% CI	*p*-Value	*ES*	95% CI	*p*-Value	*ES*
Fat mass (kg)	TG	29.37 ± 8.35	28.47 ± 8.00	0.904 ± 1.297	0.577; 1.231	<0.001	0.257	0.645; 1.193	<0.001	0.263
	CG	26.51 ± 4.84	26.36 ± 4.83	0.144 ± 0.841	−0.066; 0.035	0.175	0.009	−0.141; 0.401	0.347	0.007
Fat mass index (kg/m^2^)	TG	10.99 ± 3.32	10.71 ± 3.22	0.283 ± 0.408	0.181; 0.386	<0.001	0.221	0.195; 0.384	<0.001	0.230
	CG	10.80 ± 2.27	10.72 ± 2.24	0.080 ± 0.345	−0.006; 0.166	0.067	0.023	−0.020; 0.168	0.120	0.019
Lean mass (kg)	TG	44.15 ± 9.94	44.33 ± 9.58	−0.183 ± 1.859	−0.652; 0.285	0.437	0.008	−0.542; 0.170	0.304	0.009
	CG	37.61 ± 5.83	37.39 ± 6.23	0.219 ± 0.758	0.030; 0.408	0.024	0.012	−0.131; 0.575	0.217	0.012
Lean mass index (kg/m^2^)	TG	15.42 ± 2.27	15.53 ± 2.20	−0.114 ± 0.443	−0.226; −0.003	0.045	0.044	−0.207; −0.017	0.022	0.042
	CG	14.36 ± 1.75	14.26 ± 1.88	0.103 ± 0.301	0.028; 0.178	0.008	0.037	0.006; 0.195	0.037	0.035
MVIC right leg (N)	TG	384.16 ± 127.56	419.15 ± 135.30	−34.993 ± 48.745	−47.169; −22.816	<0.001	0.207	−45.993; −21.876	<0.001	0.199
	CG	314.27 ± 128.63	311.01 ± 116.87	3.263 ± 48.742	−8.912; 15.439	0.594	0.002	−9.854; 14.264	0.718	0.001
MVIC left leg (N)	TG	381.98 ± 132.50	422.16 ± 139.64	−40.174 ± 61.150	−55.449; −24.899	<0.001	0.154	−55.807; −22.394	<0.001	0.146
	CG	328.74 ± 126.68	306.80 ± 118.25	21.937 ± 72.785	3.756; 40.118	0.019	0.051	4.157; 37.571	0.015	0.047
MVIC arms (N)	TG	252.69 ± 99.42	306.19 ± 109.47	−53.499 ± 44.576	7.848; 20.801	<0.001	0.521	−63.370; −45.254	<0.001	0.532
	CG	226.31 ± 72.41	211.98 ± 85.46	14.325 ± 25.929	−64.726; −42.273	<0.001	0.073	6.139; 24.111	0.001	0.082
TUG (s)	TG	5.33 ± 1.04	4.65 ± 0.89	0.683 ± 0.695	0.509; 0.857	<0.001	0.393	0.533; 0.835	<0.001	0.391
	CG	5.68 ± 1.06	5.39 ± 1.09	0.291 ± 0.497	0.166; 0.415	<0.001	0.105	0.139; 0.441	<0.001	0.104
4 m gait walk (s)	TG	1.97 ± 0.40	1.89 ± 0.36	0.086 ± 0.242	0.025; 0.146	0.006	0.057	0.015; 0.135	0.014	0.047
	CG	2.07 ± 0.42	2.06 ± 0.34	0.011 ± 0.256	−0.053; 0.075	0.723	0.001	−0.038; 0.082	0.472	0.004
PH physical functioning	TG	88.75 ± 12.05	90.47 ± 11.08	−1.719 ± 7.083	−3.488; 0.051	0.057	0.011	−4.787; 1.042	0.206	0.013
	CG	90.08 ± 12.23	84.53 ± 14.60	5.547 ± 14.963	1.809; 9.285	0.004	0.102	2.786; 8.615	<0.001	0.107
PH role physical	TG	88.28 ± 23.97	94.53 ± 19.14	−6.250 ± 30.861	−13.959; 1.459	0.110	0.016	−16.163; 0.534	0.066	0.027
	CG	88.28 ± 27.08	77.34 ± 34.13	10.938 ± 38.544	1.310; 20.565	0.027	0.047	4.154; 20.850	0.004	0.066

TG: training group; CG: control group; MVIC: maximum voluntary isometric contraction; TUG: Timed Up and Go test; PH: physical health.

**Table 4 healthcare-12-00726-t004:** Inter-group differences in the pre-post change in body composition, physical fitness, and physical health variables.

			Unadjusted			Adjusted by Age		
Variable	Group	Mean Dif.	*p*-Value	*F*/*Z*	*ES*	*p*-Value	*F*/*Z*	*ES*
Fat mass (kg)	TG	−0.759	<0.001	15.368	0.109	<0.001	16.055	0.115
	CG							
Fat mass index (kg/m^2^)	TG	−0.203	0.003	9.204	0.069	0.002	9.999	0.075
	CG							
Lean mass (kg)	TG	0.402	0.112	2.561	0.020	0.115	2.514	0.020
	CG							
Lean mass index (kg/m^2^)	TG	0.217	0.002	10.481	0.077	0.002	9.662	0.073
	CG							
MVIC right leg (N)	TG	38.256	<0.001	19.711	0.135	<0.001	17.341	0.124
	CG							
MVIC left leg (N)	TG	62.111	<0.001	27.321	0.178	<0.001	24.454	0.166
	CG							
MVIC arms (N)	TG	67.824	<0.001	110.269	0.469	<0.001	113.947	0.481
	CG							
TUG (s)	TG	−0.392	<0.001	13.486	0.097	<0.001	14.288	0.104
	CG							
4 m gait walk (s)	TG	−0.074	0.094	2.846	0.022	0.219	1.528	0.012
	CG							
PH physical functioning	TG	7.266	0.001	12.327	0.089	<0.001	12.570	0.093
	CG							
PH role physical	TG	17.188	0.006	7.755	0.058	0.002	10.423	0.078
	CG							

TG: training group; CG: control group; MVIC: maximum voluntary isometric contraction; TUG: Timed Up and Go test; PH: physical health.

## Data Availability

The data presented in this study is available on request from the corresponding author. The data are not publicly available due to is personal health information.

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
