# Peer review of "The Effect of Resistance Training with Outdoor Fitness Equipment on the Body Composition, Physical Fitness, and Physical Health of Middle-Aged and Older Adults: A Randomized Controlled Trial"

_healthcare, 2024, doi:10.3390/healthcare12070726_

Round 1

Reviewer 1 Report

Comments and Suggestions for Authors

From a methodological standpoint, the research is, in my opinion, impeccable. It has a correct design, and the CONSORT protocol is applied, which controls this type of clinical trial. Therefore, summarizing, I congratulate the authors for their mastery of research techniques.

Having said all this, I regret to inform you that I cannot recommend the publication of the paper. The reason is as follows:

The effect of strength training on individuals aged 50 and older is widely known. As the authors are undoubtedly aware, the initial guidelines on physical activity recommend at least two days a week of resistance exercises. The scientific evidence is overwhelming and demonstrates that it improves physical fitness, objective health, perceived health, as well as quality of life in its physical, psychological, and social dimensions. In summary, the scientific evidence is at its maximum (Type A). In 2008, a group of specialists, led by Professor Chodzko-Zajko from the University of Illinois at Urbana Champaign, conducted an interesting review on these topics, which established the position at that time of the American College of Sports Medicine. Likewise, this is reflected in the World Health Organization's guidelines on physical activity.

Therefore, in my opinion, I do not consider the results stated in the paper's conclusions to be impactful, which essentially state: "The present study found that resistance training with OFE, conducted twice a week for 8 weeks, resulted in significant improvements in body composition, physical fitness, health, and quality of life among middle-aged and older adults compared to a control group that continued with their usual daily activities. These findings are of considerable interest for the clinical field, healthcare professionals, and the sports industry, particularly for strength trainers. Within this article, you will find the entire training program developed, allowing for direct practical application."

The impact of these conclusions is minimal, as the hypothesis of the effects of resistance training on the variables indicated in the research is widely supported.

Author Response

Reviewer 1

From a methodological standpoint, the research is, in my opinion, impeccable. It has a correct design, and the CONSORT protocol is applied, which controls this type of clinical trial. Therefore, summarizing, I congratulate the authors for their mastery of research techniques.

  • Thank you very much. We are glad that you found the article interesting. We have done the revisions indicated. We believe that the current version has improved on the previous version. We remain at your disposal for any future questions you may have.

Having said all this, I regret to inform you that I cannot recommend the publication of the paper. The reason is as follows: The effect of strength training on individuals aged 50 and older is widely known. As the authors are undoubtedly aware, the initial guidelines on physical activity recommend at least two days a week of resistance exercises. The scientific evidence is overwhelming and demonstrates that it improves physical fitness, objective health, perceived health, as well as quality of life in its physical, psychological, and social dimensions. In summary, the scientific evidence is at its maximum (Type A). In 2008, a group of specialists, led by Professor Chodzko-Zajko from the University of Illinois at Urbana Champaign, conducted an interesting review on these topics, which established the position at that time of the American College of Sports Medicine. Likewise, this is reflected in the World Health Organization's guidelines on physical activity.

  • Thank you for your comment. You are right, and the present research is not intended to find out whether strength training is effective for older people, as this has been proven to be the case. The aim is to show that these adaptations can also be achieved with an outdoor fitness equipment circuit, even if it only allows self-loading. A substantial part of the introduction has been modified to make the gap leading to the present research clearer.

Therefore, in my opinion, I do not consider the results stated in the paper's conclusions to be impactful, which essentially state: "The present study found that resistance training with OFE, conducted twice a week for 8 weeks, resulted in significant improvements in body composition, physical fitness, health, and quality of life among middle-aged and older adults compared to a control group that continued with their usual daily activities. These findings are of considerable interest for the clinical field, healthcare professionals, and the sports industry, particularly for strength trainers. Within this article, you will find the entire training program developed, allowing for direct practical application." The impact of these conclusions is minimal, as the hypothesis of the effects of resistance training on the variables indicated in the research is widely supported.

  • Thank you for your comments. The conclusion has been reworded to specify what is really new about this research and the impact of this work.

Reviewer 2 Report

Comments and Suggestions for Authors

The manuscript is well-written in an engaging and lively style. It’s currently something of a “stormy topic” and is one to which the author has made significant contributions in sports training and sports science field. The purpose of the manuscript was to examine the effect of outdoor fitness equipment-based resistance training on health parameters of middle-aged and older adults. This study seem the extension or extraction of the study published (https://bmjopensem.bmj.com/content/9/4/e001829) in the journal BMJ open sport & Exercise Medicine. Some areas require rewriting or clarification. I comment on these areas section by section in the author's column.

Title of the article: It is mentioned that resistance training affects on middle-age and older adults.

However, there was no separate data for middle-aged and older adults.

And no comparison was made between middle-aged and older adults.

Abstract and introduction:

The abstract and introduction were written in a good form.

Materials and Methods

Experimental Approach:  How was the randomization done?

Subjects: Why anthropometric data was not present as height, weight, and BMI.

What is the significance of the blindness of the examiner and staff who performed the statistical analysis? This is not defined in CONSORT guidelines.

Table 2 should be elaborated in one paragraph for clear understanding.

L 131- does the researcher find any difficulty in bringing participants from private and reserved study areas within the facilities of the Catholic University of Murcia (UCAM) to the sports science laboratory to perform measurements?

L 133 Where is the data on physical activity level, illness and injuries? It might have a significant effect on the outcome measures.

Statistical analysis (Sample size): As per the information mentioned in the L 161-162 as standard deviation (4.2) in the maximum voluntary isometric contraction (MVIC)

The standard deviation (4.2) is not available in the study [27] for maximum voluntary isometric contraction.
Authors should need to calculate the correct sample size again.  

Results:

L 189-194 Table 4 shows the differences between the TG and CG in the pre-post change.

This information is mystifying because either this information is the result of a comparison between TG and CG in only one stage pre- or post,

Or table shows the difference between the pre and post-test of one group.

From Table 4, it cannot be concluded that significant increase or decrease.

The authors should provide the interpretation of the calculated effect size.  

The strength of the study should be provided.

Author Response

The manuscript is well-written in an engaging and lively style. It’s currently something of a “stormy topic” and is one to which the author has made significant contributions in sports training and sports science field. The purpose of the manuscript was to examine the effect of outdoor fitness equipment-based resistance training on health parameters of middle-aged and older adults.

  • Thank you very much. We are glad that you found the article interesting.

This study seem the extension or extraction of the study published (https://bmjopensem.bmj.com/content/9/4/e001829) in the journal BMJ open sport & Exercise Medicine.

  • Yes, it is. The article you indicate is the protocol of the present investigation. This has been clarified in the paper.

Some areas require rewriting or clarification. I comment on these areas section by section in the author's column.

  • We have covered the revisions indicated. We believe that the current version has improved on the previous version. We remain at your disposal for any future questions you may have.

Title of the article: It is mentioned that resistance training affects on middle-age and older adults. However, there was no separate data for middle-aged and older adults. And no comparison was made between middle-aged and older adults.

  • Thank you for your excellent comment. Regarding the differentiation between middle-aged and older adults in our study, we have taken your observation into consideration and have incorporated an age-adjusted analysis into our statistical analysis to determine if this variable could influence the outcomes. With this information, the sections on abstract, objective, statistical analysis, results, discussion, limitations and conclusions have been modified. We believe that this adjustment will enhance the findings of our study, allowing us to provide more specific and relevant conclusions.

Abstract and introduction: The abstract and introduction were written in a good form.

  • Thank you for your comment.

Materials and Methods: Experimental Approach:  How was the randomization done?

  • Thank you for your comment. We have included a new section in the manuscript detailing this process. This section explains the methods we used for randomizing participants into different study groups to ensure the integrity and reliability of our findings.

Subjects: Why anthropometric data was not present as height, weight, and BMI.

  • Thank you for your comment and I apologise for this error. Information about the height, body mass, and BMI of the participants has been included.

What is the significance of the blindness of the examiner and staff who performed the statistical analysis? This is not defined in CONSORT guidelines.

  • Thank you for your comment. A new section has been included including detailed information on the blinding of the study.

Table 2 should be elaborated in one paragraph for clear understanding.

  • Thank you for your comment. A new paragraph has been included, explaining Table 2 in detail to enhance understanding.

L 131- does the researcher find any difficulty in bringing participants from private and reserved study areas within the facilities of the Catholic University of Murcia (UCAM) to the sports science laboratory to perform measurements?

  • Thank you for your comment. Sorry if this was not clear in the previous version, this section has been rewritten for clarity.

L 133 Where is the data on physical activity level, illness, and injuries? It might have a significant effect on the outcome measures.

  • Thank you for your comment. Information on physical activity level, illness and injuries has been included, and your comment has been considered as a possible limitation of the study.

Statistical analysis (Sample size): As per the information mentioned in the L 161-162 as standard deviation (4.2) in the maximum voluntary isometric contraction (MVIC). The standard deviation (4.2) is not available in the study [27] for maximum voluntary isometric contraction. Authors should need to calculate the correct sample size again. 

  • Thank you for your comment. We apologize for the error mentioned regarding the standard deviation in the maximum voluntary isometric contraction (MVIC) not being available in study [27]. This oversight has been corrected by recalculating the sample size based on a similar study with appropriate characteristics.

Results: L 189-194 Table 4 shows the differences between the TG and CG in the pre-post change. This information is mystifying because either this information is the result of a comparison between TG and CG in only one stage pre- or post, Or table shows the difference between the pre and post-test of one group. From Table 4, it cannot be concluded that significant increase or decrease.

  • Dear reviewer. We thank you for this accurate comment and we are sorry for the error. The title of table 4 and the paragraph in the text have been changed to adjust the interpretation to what has actually been done.

The authors should provide the interpretation of the calculated effect size.

  • Thank you for your comment. The interpretation of the calculated effect size has been included in the current version.

The strength of the study should be provided.

  • Thank you for your comment. A paragraph has been included at the end of the discussion with this information.

Reviewer 3 Report

Comments and Suggestions for Authors

The authors are commended for their comprehensive study, which encompasses all requisite components for approval.

 To succinctly encapsulate, my observations and critiques are inherently subjective. Thus, suggesting further alterations to the study would likely yield negligible benefits.

 However, it is noteworthy that the study lacks significant innovative contributions. Despite the authors published a literature review on the subject (10.1016/j.exger.2023.112279), others have previously disseminated similar articles in alternative publications, potentially compromising novelty (pleas refers to academic google or PubMed using the terms “outdoor fitness equipment older adults”). Additionally, the adequacy of exercise control remains ambiguous, particularly regarding the regulation of movement velocity, which can significantly impact training intensity. Lastly, the omission of equipment replication as a limitation warrants attention. Ensuring uniformity in equipment configuration across different settings worldwide is crucial. Consequently, providing such information is imperative for comprehensive evaluation.

Author Response

The authors are commended for their comprehensive study, which encompasses all requisite components for approval.

  • Thank you very much. We are glad that you found the well structured.

To succinctly encapsulate, my observations and critiques are inherently subjective. Thus, suggesting further alterations to the study would likely yield negligible benefits.

  • Thank you very much for your comments. We have covered the revisions indicated. We believe that the current version has improved on the previous version. We remain at your disposal for any future questions you may have.

However, it is noteworthy that the study lacks significant innovative contributions. Despite the authors published a literature review on the subject (10.1016/j.exger.2023.112279), others have previously disseminated similar articles in alternative publications, potentially compromising novelty (pleas refers to academic google or PubMed using the terms “outdoor fitness equipment older adults”).

  • Thank you for your comment. In the revised version of the document, we have emphasized the unique contributions of our study, including a detailed justification of its relevance. We focus on the distinctive characteristics of the outdoor fitness equipment and our specific intervention protocol and methodology, which together address a gap in the existing literature on resistance training for middle-aged and older adults. We believe these aspects of our research contribute valuable new knowledge to the field, enhancing the understanding of how different approaches to resistance training can benefit this population.

Additionally, the adequacy of exercise control remains ambiguous, particularly regarding the regulation of movement velocity, which can significantly impact training intensity.

  • Thank you for your comment. Sorry if this was not clear in the previous version, this section has been rewritten for clarity, emphasizing that the movement is constant and controlled and is supervised by the trainers.

Lastly, the omission of equipment replication as a limitation warrants attention. Ensuring uniformity in equipment configuration across different settings worldwide is crucial. Consequently, providing such information is imperative for comprehensive evaluation.

  • Thank you, your comment has been included as a possible limitation of the study concluding that it is important to note that if this training is not conducted with the exact machines and set up used in this study, the results may not be applicable.

Round 2

Reviewer 1 Report

Comments and Suggestions for Authors

I thank the authors for making the modifications they suggested to the paper. Consider that it is now clear in the introduction what the research is intended to do. The authors have focused the research objective much better.

On the other hand, the reformulation of the conclusions is now in line with the objective of the document.

In my opinion the contribution to scientific knowledge of this article is low. In any case, I am going to recommend its publication, as it adds more evidence about training in older people, and considers that this is a target objective, given the aging of the population.

Reviewer 2 Report

Comments and Suggestions for Authors

The authors satisfactorily addressed all the comments on the manuscript. 

I recommend that this manuscript be accepted for further processing.